# Meteorological and Agricultural Drought Risk Assessment via Kaplan–Meier Survivability Estimator

**Cem Polat Cetinkaya \***  **and Mert Can Gunacti**

Department of Civil Engineering, Faculty of Engineering, Tinaztepe Campus, Dokuz Eylül University, Buca, Izmir 35220, Turkey; mert.gunacti@deu.edu.tr
* Correspondence: cem.cetinkaya@deu.edu.tr

**Abstract:** Dry periods and drought are inherent natural occurrences. However, due to the increasing pressures of global warming and climate change, these events have become more frequent and severe on a global scale. These phenomena can be traced with various indicators and related indices proposed by various scholars. In general, drought risk assessment is done by modeling these indicators and determining the drought occurrence probabilities. The proposed adaptation introduces the "Kaplan–Meier estimator", a non-parametric statistic traditionally used in medical contexts to estimate survival functions from lifetime data. The study aims to apply this methodology to assess drought risk by treating past droughts as "events" and using drought indicators such as the Standardized Precipitation Index (SPI) and Standardized Precipitation Evapotranspiration Index (SPEI). Mapping these results for a better understanding of the drought risks on larger spatial scales such as a river basin is also within the expected outcomes. The adapted method provides the probability of non-occurrence, with inverted results indicating the likelihood of drought occurrence. As a case study, the method is applied to *SPI* and *SPEI* values at different time steps (3, 6, and 12 months) across 27 meteorological stations in the Gediz River Basin, located in Western Turkey—a region anticipated to be profoundly affected by global climate change. The results are represented as the generated drought risk maps and curves, which indicate that (i) drought risks increase as the considered period extends, (ii) drought risks decrease as the utilized indicator timescales increase, (iii) locally plotted drought curves indicate higher drought risks as their initial slope gets steeper. The method used enables the generation of historical evidence based spatially distributed drought risk maps, which expose more vulnerable areas within the river basin.

**Keywords:** Kaplan–Meier estimator; drought risk assessment; SPI; SPEI



## 1. Introduction

Drought, a recurring and multifaceted phenomenon, is a widely studied topic that is a complex interplay between atmospheric, land, and water resource systems [1–3]. It manifests through prolonged periods of below-average precipitation, leading to depleted soil moisture, reduced surface water availability, and cascading impacts across various sectors [4]. Its negative impacts are far-reaching, including agricultural losses, water and food insecurity, economic instability, mass migrations, and environmental degradation [5].

Particularly, the effects of drought on agriculture impact economic, social, and environmental aspects on several levels, because the agricultural sector is the first sector affected by drought, as a drought may reduce water availability in the soil, increase crop failure and pasture losses, reduce crop yield, and threaten food security [6]. While meteorological droughts are associated with precipitation deficits, agricultural droughts are associated with deficits in soil moisture [7].

In order to develop strategies and approaches to mitigate the effects of drought and provide water security, firstly the concept of drought in the examined case study is defined in its current state [5,8,9]. This is expressed by drought indicators based on the historical

timeseries of water sources, such as precipitation, streamflow, etc. Drought indicators help us to define quantitative or qualitative parameters of the system for the evaluation of drought monitoring and prediction [10,11]. These tools may then eventually serve decision-makers in developing policies against the adverse effects of droughts.

The study uses two of the most widely used drought indicators—the Standardized Precipitation Index (SPI) [12] and its climatic water balance variant, the Standardized Precipitation–Evapotranspiration Index (SPEI) [13]—to evaluate the droughts experienced in the study area. Due to their low data requirements, ease of calculation and interpretation, and flexible natures, *SPI* and *SPEI* have been a staple for drought studies [14–18].

While the drought indicators may define the states of the hydrologic systems, drought risk assessment is the crucial step for informed decision-making [19], targeting drought interventions [20], enhanced preparedness [21], the resilience of the hydrologic systems [22], risk reduction and cost–benefit analysis [23], and sustainable development and climate change adaptation [24]. However, defining drought risk is an abundantly discussed topic, as there are newly introduced concepts such as critical drought severity, singular drought, and within-period drought [25]. Although these discussions are useful for our understanding of the complex concepts of drought and its risk, the presented study offers a more practical method that requires fewer data for the determination of drought risk, which is important considering low data availability on the global scale.

In that regard, the Kaplan–Meier estimator has been used in the evaluation of the *SPI/SPEI* drought indicators; it was originally introduced by Kaplan and Meier in 1958 [26] and is a widely used method for analyzing survival data in medical science, which is the calculated value of human patients' survivability probability after a certain treatment. The Kaplan–Meier estimator's popularity stems from its key attributes, such as its nonparametric nature, its direct translation of observed data into a survival curve, censored data handling, and statistical robustness. Although parametric methods have been deemed to be more robust than the nonparametric ones, their ease of rapid calculation and low input needs make them essential for this study [27]. The study uses the Kaplan–Meier estimator based on the constructive interaction whereby every wet period ends with a following dry period, which can be considered as the limit of the wet period. Determining the survival curve of the study area based on the *SPI* and *SPEI* values would then indicate the survival of the wet periods. An inverse calculation defines the survival probabilities of the dry periods, or simply the drought risks.

The assessment of drought risk entails a variety of approaches and applications around the globe, where researchers usually monitor drought risk as a combination of some sub-indicators both around the globe [28–31] and within national boundaries [32–35]. The study supports these approaches, as the adapted methodology can be used with other drought indicators.

This study's objective is to explore the drought risks associated with the data from meteorological stations within the designated study area, by extrapolating the spatial distribution between these stations to generate comprehensive drought risk maps across different periods (Figure 1). The main difference between the approach of drought assessment through *SPI/SPEI* and the newly introduced methodology is that the indices SPI and *SPEI* determine the dry/wet states of the examined location based on precipitation plus the *PET* for the available period, while the newly proposed methodology defines the occurrence risk of a drought after a wet period using the Kaplan–Meier estimator through the use of the drought indicators of *SPI* and *SPEI* in this study. Furthermore, the Kaplan–Meier estimator can also be used for different indices, such as the Reconnaissance Drought Index (RDI) [36], the Streamflow Drought Index (SDI) [37], and the Precipitation Deciles (PD) [38]. This research introduces an innovative approach to assessing drought risks within the examined region, aiming to assist decision-makers in prioritizing, evaluating, and formulating relevant policies.

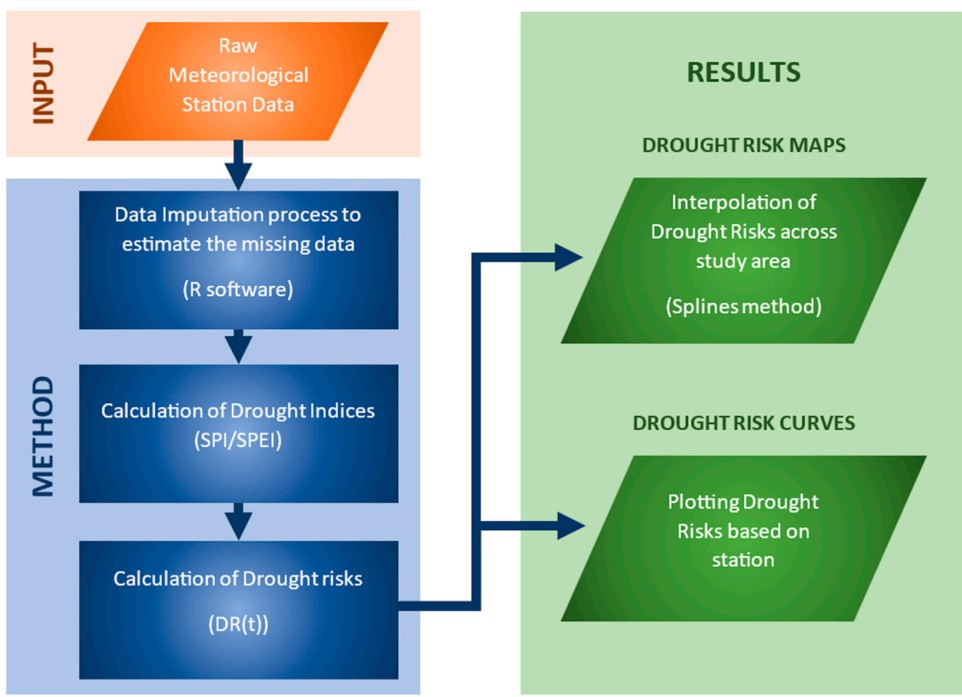

**Figure 1.** Flowchart of the study.

Conversely, it is crucial to acknowledge certain limitations of the study, that primarily revolve around the availability of temporal and spatial data, as well as the precision and accuracy of extrapolated drought risk values due to the resolution of data representation.

## 2. Materials and Methods

### 2.1. Study Area and Data Selection

The study area is the Gediz River Basin (GRB) located in Western Turkey. The average annual precipitation is 603 mm (ranging from 500 mm near the Aegean coast to 1000 mm in the mountainous eastern end), and the average annual temperature ranges between 12.3 and 16.8 °C in the basin [39]. According to the Köppen–Geiger classification, the basin is classified as "Csa", which is described as "Hot-summer Mediterrenean climate" [40]. The basin drastically changed in the 1990s, from a water-rich one to one that struggles with water scarcity. The frequent meteorological and hydrological droughts and increase in urban and industrial demands are the main drivers of this change [41]. The agricultural activities in the basin have also been affected by these changes. The distribution of the main crop types cultivated in the basin (cotton, maize, and grapes) changed over time to adapt to the water scarcity, and the total irrigated area shrunk to half its potential size by the year 2012 due to maintenance operation problems and a lack of surface water [42].

The basin has over sixty meteorological stations in and around its boundaries; however, some of the stations lack reliable data as some others are too close to each other, causing repeated information (Figure 2). Data availability, a common challenge globally, is one of the primary driving factors in choosing the best stations to represent the study area in time and space dimensions. Out of 68 meteorological stations considered around GRB, 27 were selected according to their data availability and spatial location, aiming at homogeneity in data representation. The basin substantially changed in the 1990s due to anthropogenic activities and climate change-related pressures. Thus, the declining water resources and increasing urban and industrial demands raise concerns regarding water quality and quantity. Determining and projecting potential future droughts is a valuable tool in decision-making and planning for the basin [43].

The station records range between the years 1924 and 2013, but the majority of the data are available for between the 1960s and the late 1990s (Table S1). The missing data from

the selected stations were imputed by the R software (version 4.2.2) according to linear regression. The R software is a free and open-source programming language and software environment for statistical computing and graphics. The package "MICE" was used in the data imputation process [44]. After the data imputation process, completed continuous timeseries were used for the *SPI/SPEI* calculation process. Since the main analysis of drought indicators for this stage is a localized one, based on the total available data of each station, the consideration of a common period is not a necessity, as the occurrence of a drought with any intensity after a wet period is the focus of this study.

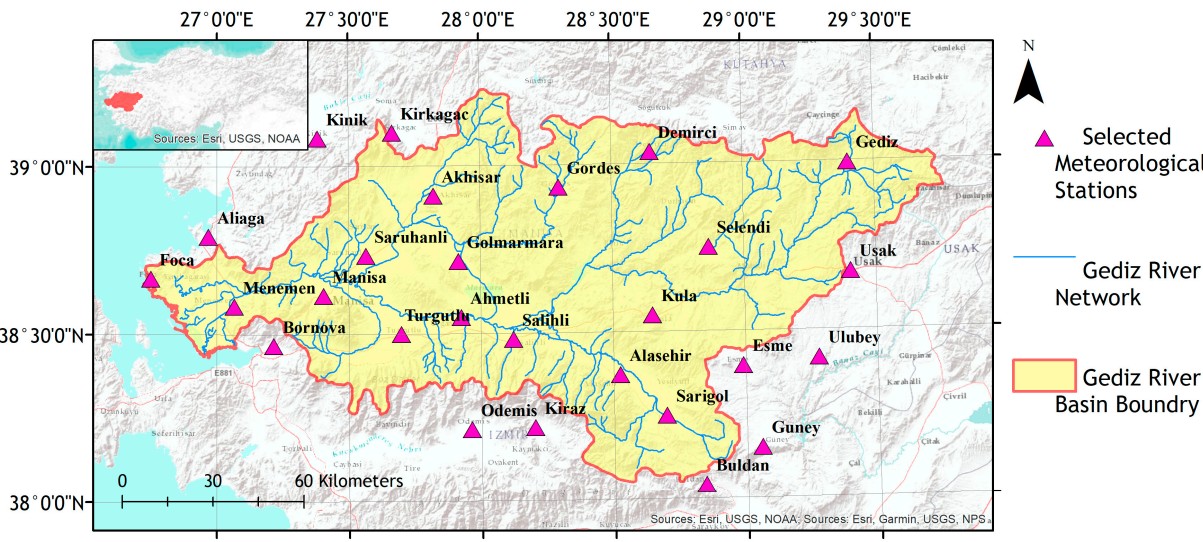

**Figure 2.** Gediz River Basin and selected meteorological stations.

### 2.2. SPI/SPEI Calculations

The *SPI* is computed by the cumulative precipitation over k months, defined as the accumulation periods, fitted to a parametric statistical distribution from which probabilities are transformed to the standard normal distribution. The *SPEI* is determined in the same manner, with a difference in that the cumulative climatic water balance is defined as the difference between precipitation and the Potential Evapotranspiration (PET). The PET described in this study has been calculated according to the Hargreaves equation [45], which actually calculates reference evapotranspiration ($ET_0$) but is considered equivalent [46]. The determined probabilities are then converted to the standard normal distribution to generate the final drought index values. Equations for *SPI* and *SPEI* are given below,

$$SPI = \frac{P_i - P_m}{\sigma_x} \tag{1}$$

$$SPEI = \frac{D_i - D_m}{\sigma_d} \tag{2}$$

where $P_i$ is the total precipitation of the station on the *i*-th month, $P_m$ is the mean precipitation, $\sigma_x$ is the standard deviation of the analyzed precipitation series, $D_i$ is the difference between precipitation (P) and *PET* for the month (*i*), $D_m$ is the mean difference between *P* and *PET*, and $\sigma_d$ is the standard deviation of the analyzed "D" difference timeseries. For the calculation of *SPEI*, ($D_i$) is given in Equation (3)

$$D_i = P_i - PET_i \tag{3}$$

The calculated D values are accumulated at different time scales, as follows:

$$D_n^k = \sum_{i-0}^{k-1} P_{n-1} - (PET)_{n-1} \tag{4}$$

where $k$ is the timescale (months) of the aggregation and $n$ is the calculation month.

The probability density function of a log-log distribution is given as:

$$f(x) = \frac{\beta}{\alpha} \left( \frac{x - \gamma}{\alpha} \right)^{\beta - 1} \left( 1 + \left( \frac{x - \gamma}{\alpha} \right)^{\beta} \right)^{-2} \tag{5}$$

where $\alpha$, $\beta$ and $\gamma$ are scale, shape, and origin parameters, respectively, for $\gamma > D < \infty$. The probability distribution function for the $D$ series is then given as:

$$f(x) = \left[ 1 + \left( \frac{\alpha}{x} - y \right)^{\beta} \right]^{-1} \tag{6}$$

With $f(x)$ the *SPEI* can be obtained as the standardized values of $F(x)$ according to the method of Abramowitz et al. (1965) [47]:

$$SPEI = W - \frac{C_0 + C_1 W + C_2 W^2}{1 + d_1 W + d_2 W^2 + d_3 W^3} \tag{7}$$

$$W = \sqrt{-2 \ln(P)} \quad for \ P \leq 0.5 \tag{8}$$

where $P$ is the probability of exceeding a determined $D_i$ value and is given as $P = 1 - f(x)$, while the constants are:

$$C_0 = 2.515517, \ C_1 = 0.802853, \ C_2 = 0.010328,$$

$$d_1 = 1.432788, \ d_2 = 0.189269, \ d_3 = 0.001308$$

The *SPI* and *SPEI* values for the 3-, 6-, and 12-month accumulation periods were calculated for the selected stations by the "SPEI package" of the R software. Shorter timescales such as 3, 6, and 12 months are generally used as indicators for reduced soil moisture and flow in relatively small tributaries of a river, and longer timescales are commonly used as indicators for reduced streamflow and reservoir storage. They can be used as indicators of different types of droughts.

*2.3. Adaption and Adoption of Kaplan–Meier Estimator for Drought Risk Assessment*

The Kaplan–Meier estimator methodology is used to analyze "time-to-event" data, usually in medical sciences. In the term "Time-to-event", the term "event" is considered as the "death" of a patient after some applied treatment in medical science, and provides researchers the flexibility to apply this method to other fields by altering the event term to apply to the fatigue strength of metals [48] or the employment of married women [49]. The method's main assumptions regard the censored data, which are (i) at any time, patients who are censored have the same survival prospects as those who continue to be followed; (ii) the survival probabilities are the same for subjects recruited early and late in the study, (iii) the event happens at the time specified, none of which is related to the presented study since they are mostly related to patients' human behavior in relation to joining or leaving the study. Thus, due to the nature of the study, the term "product limit" describes the process of the study better.

The calculated values of *SPI* and *SPEI* are classified as wet and dry states (Table 1). The transition of wet state to dry state or drought occurrence indicates the "events" described by Kaplan–Meier [26]. The study is based on this synergy, where drought occurrences are named "events". Thus, applying the Kaplan–Meier or "Product-Limit" estimator would produce the product limit or the survivability of the wet periods (in this case, in months).

$$\hat{S}(t) = \prod_{i: \ t_i \leq t} \left( 1 - \frac{d_i}{n_i} \right) \tag{9}$$

where $t_i$ is the time at which a dry state occurs; $n_i$ is the total number of wet states at the time $t_i$ and $d_i$ is the number of dry states that occurred at time $t_i$.

Since dry states can occur consecutively, the adapted method also includes the following dry state occurrences in the same wet period. This provides additional knowledge about the occurrence probabilities of the following dry states, not just the initial one, e.g., Ahmetli station has 32 wet periods but 44 recorded dry states. Including every dry state within the equation requires an update of the description of $n_i$, which now can be explained as the "number of dry states hasn't happened yet at the time $t_i$" or the "number of unrealized dry states".

**Table 1.** *SPI/SPEI classification scale.*

| SPI/SPEI Values | Description of State |
|---|---|
| SPI/SPEI < −2 | Extreme drought |
| −2 < SPI/SPEI < −1.5 | Severe drought |
| −1.5 < SPI/SPEI < −1 | Moderate drought |
| −1 < SPI/SPEI < 1 | Near normal |
| 1 < SPI/SPEI < 1.5 | Moderately wet |
| 1.5 < SPI/SPEI < 2 | Severely wet |
| SPI/SPEI > 2 | Extremely wet |

The result $S(t_i)$ indicates the probability of a wet period lasting longer than $t_i$ or the "event" not taking place at the time $t_i$; in other words, the nonconcurrence probability of the drought. From the drought risk perspective, $1 - S(t)$ would describe the probability of an "event" taking place or the occurrence probability, which is the definition of risk, $DR(t)$ (Table 2).

$$DR(t) = 1 - \hat{S}(t) = 1 - \left( \prod_{i:\, t_i \leq t} \left( 1 - \frac{d_i}{n_i} \right) \right) \tag{10}$$

**Table 2.** Recorded dry states in Ahmetli station between 1966 and 1988 according to *SPI*-3.

| Time (Month) | No. Unrealized Dry States $r_i$ | No. Dry States Realized $d_i$ | Kaplan–Meier $\hat{S}(t)$ | Drought Risk $(DR(t))$ |
|---|---|---|---|---|
| 1 | 44 | 5 | 0.886 | 0.114 |
| 2 | 39 | 5 | 0.773 | 0.227 |
| 3 | 34 | 6 | 0.636 | 0.364 |
| 4 | 28 | 6 | 0.500 | 0.500 |
| 5 | 22 | 3 | 0.432 | 0.568 |
| 6 | 19 | 2 | 0.386 | 0.614 |
| 7 | 17 | 2 | 0.341 | 0.659 |
| 8 | 15 | 4 | 0.250 | 0.750 |
| 9 | 11 | 2 | 0.205 | 0.795 |
| 10 | 9 | 2 | 0.159 | 0.841 |
| 14 | 7 | 1 | 0.136 | 0.864 |
| 15 | 6 | 1 | 0.114 | 0.886 |
| 17 | 5 | 1 | 0.091 | 0.909 |
| 18 | 4 | 1 | 0.068 | 0.932 |
| 24 | 3 | 1 | 0.046 | 0.954 |
| 25 | 2 | 1 | 0.023 | 0.977 |
| 27 | 1 | 1 | 0 | 1.000 |

In the first column of Table 2, the observed durations (months) that a dry state occurs after a wet period are given for the specific station, e.g., the longest duration for which a dry state occurrence was observed after a wet state is 27 months for Ahmetli station. In

the next column, the number of unrealized dry states, which is the cumulative number of dry state occurrences (events), is listed in decreasing order. The number of realized dry states $d_i$ describes the number of dry state occurrences of each observed duration. The Kaplan–Meier survival probability is calculated in the $\hat{S}(t)$ column and the drought risks are determined in the last column.

## 3. Results and Discussion

### 3.1. Drought Risk Map Results

The drought risks of the 27 meteorological stations, according to the 3-, 6-, and 12-month *SPI/SPEI* values and the adapted Kaplan–Meier estimator method, are here calculated (Tables S2–S217). According to the results, 3-, 6-, and 12-month drought risk values for each station have been integrated with the stations' locations via ArcGIS software (version 10.3.1). Local *SPI/SPEI* drought risk values were then plotted as drought risk maps using ArcGIS. Point values have been extrapolated via the "SPLINE" tool of ArcGIS (Figures 3–8), which is based on the splines methodology described by Wahba [50] and was computationally developed by Hutchinson [51].

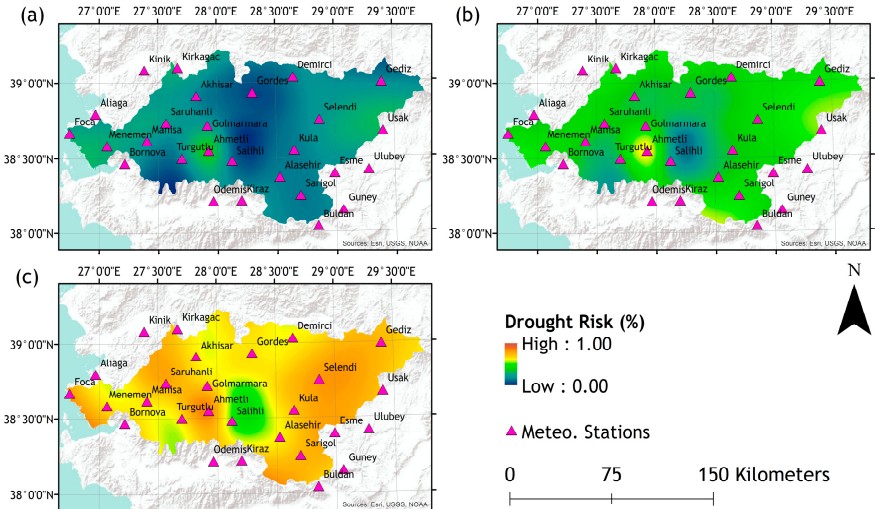

**Figure 3.** Drought risk maps of (**a**) 3-month, (**b**) 6-month, and (**c**) 12-month periods according to *SPI*-3.

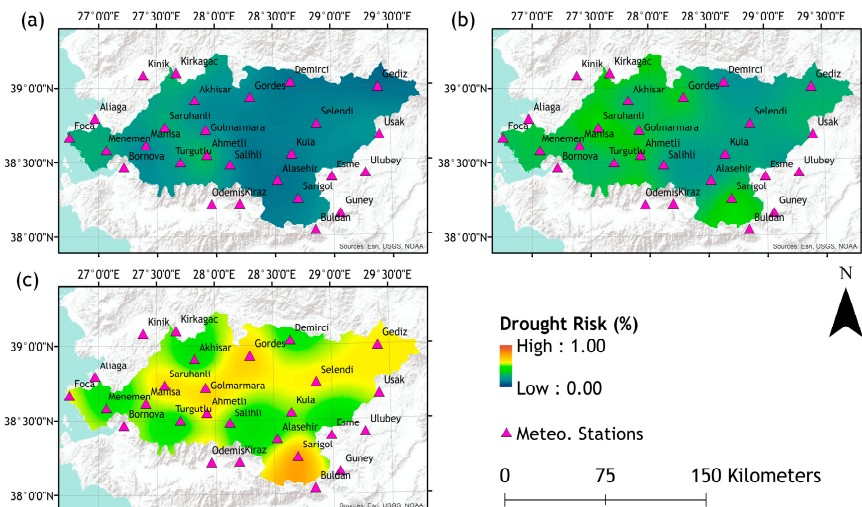

**Figure 4.** Drought risk maps of (**a**) 3-month, (**b**) 6-month, and (**c**) 12-month periods according to *SPI*-6.

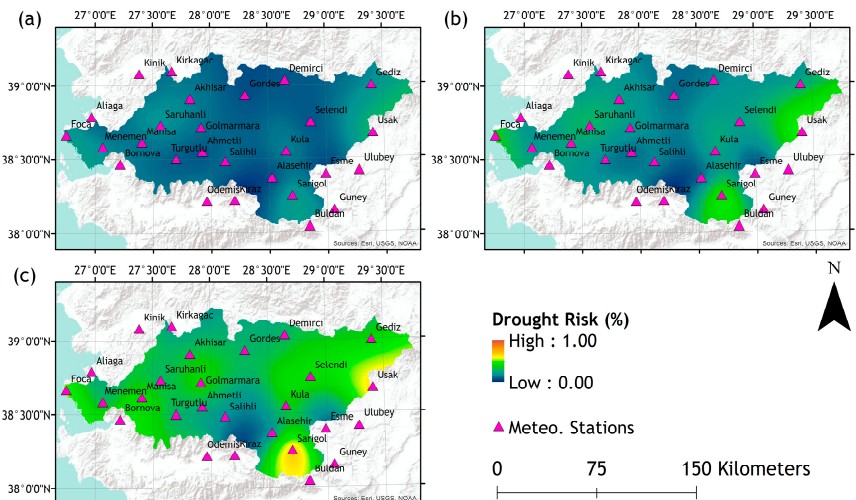

**Figure 5.** Drought risk maps of (**a**) 3-month, (**b**) 6-month, and (**c**) 12-month periods according to *SPI*-12.

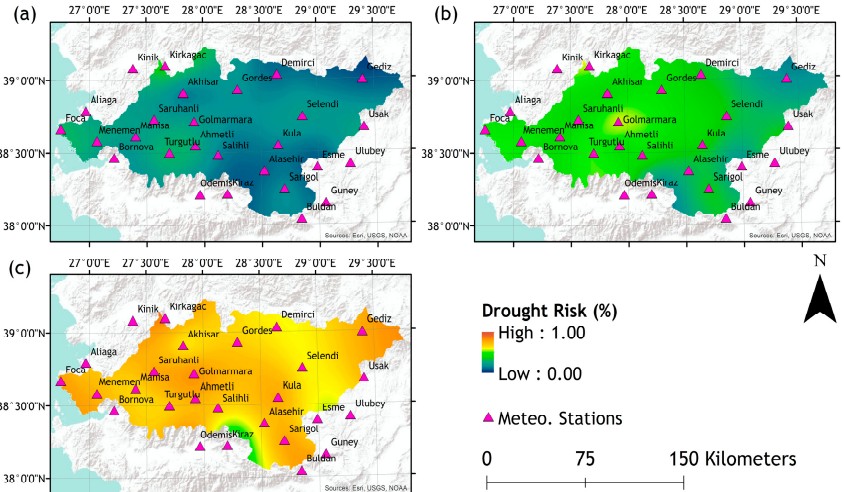

**Figure 6.** Drought risk maps of (**a**) 3-month, (**b**) 6-month, and (**c**) 12-month periods according to *SPEI*-3.

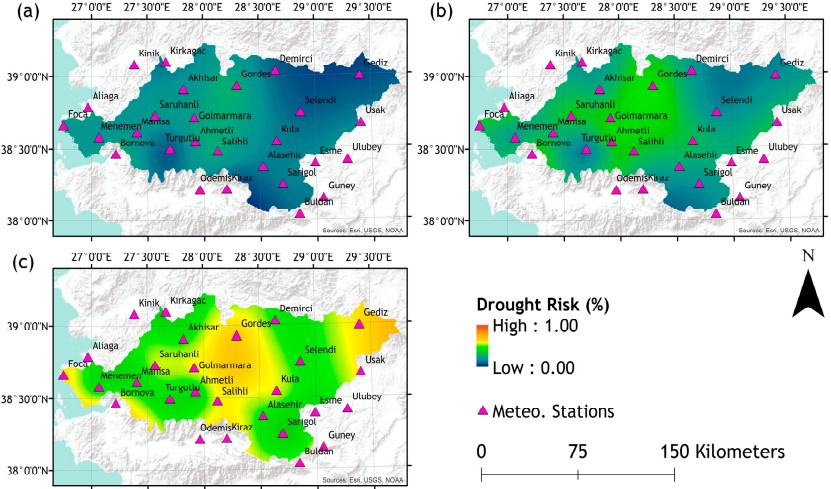

**Figure 7.** Drought risk maps of (**a**) 3-month, (**b**) 6-month, and (**c**) 12-month periods according to *SPEI*-6.

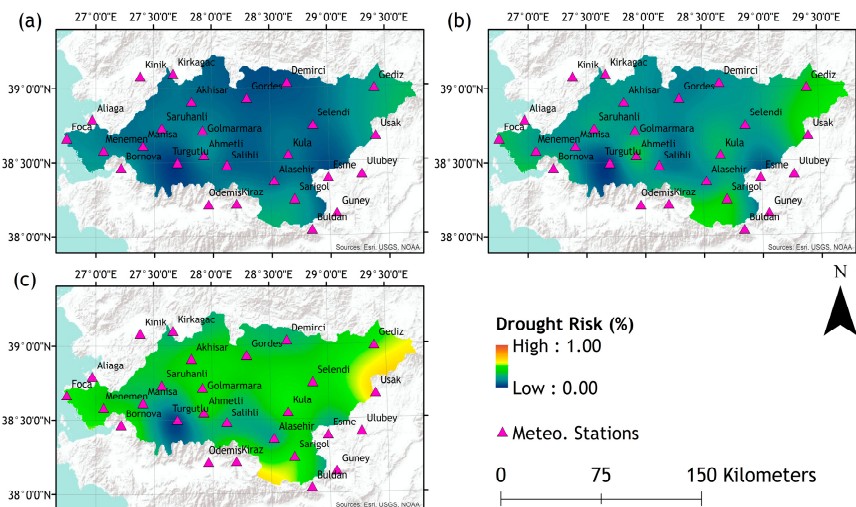

**Figure 8.** Drought risk maps of (**a**) 3-month, (**b**) 6-month, and (**c**) 12-month periods according to *SPEI*-12.

The plotted drought risk maps present the drought risk possibility after a 3-, 6-, or 12-month period according to *SPI* or *SPEI* values of 3-, 6-, or 12-month timescales (Figures 3–8). While the lower drought risk percentages are represented by colder color tones (blue-green), higher risk percentages are represented by warmer color tones (red-orange). As demonstrated, this array of visual results can be used as tools for decision-making by various levels of decision-makers and stakeholders. As the density of information given to end-users and decision-makers may be confusing due to their complex nature, spatially distributed maps are helpful tools that help to demonstrate vulnerable hotspots for different time scales to aid with the comprehension of drought risks by stakeholders and decision-makers. It must be noted that the drought risk maps are generated based on a specific case; for instance, a request for drought risk mapping for the next cultivation season will require a shorter time scale of a chosen drought index, such as *SPI*, but when the presentation of the climate change effect is a necessity, another drought risk mapping approach based on *SPEI* for a longer planning activity may be preferred by a decision-maker responsible the implementation of policies and preventive measures.

The drought risk maps produced can be interpreted both via comparisons and via point value. For instance, in Figure 3, Ahmetli station represents higher drought risks compared to the other locations around it, but specifically, according to the *SPI-3*, there are 36%, 61% and 85% risks that Ahmetli will experience drought in the next 3, 6 and 12 months, respectively.

In general, each of the results plotted and shown in Figures 3–8 indicate that, as the period of the drought risk assessed increases, the drought risk also increases. This is due to the fact that as the inquiry period extends, it is more likely that the months considered as "dry" will also be included in the calculations. It is also observed that as the timescales of the calculated *SPI* or *SPEI* values increase, drought risks decrease. As the timescales of the *SPI* or *SPEI* increases, the accumulated values tend to reach an average, and thus the most extreme events such as floods or droughts are inclined to diminish.

Conversely, a comparison between the *SPI* and *SPEI* drought risk maps indicates that, for most of the study area, the observed results are similar, aside from slight differences due to information on *PET* added by the *SPEI*.

### 3.2. Drought Risk Curve Results

While drought risk maps help to capture the general outline and condition of the study area, drought risk curves can also be used to focus on specific locations, such as Lake Marmara in the Gediz study case. The drought risk curves of Lake Marmara, derived from *SPI* (Figure 9a) and *SPEI* (Figure 9c) values, validate the historical records of regional

droughts experienced by local farmers first-hand. Situated in the center of the GRB, Lake Marmara suffers severely from droughts, and drought risks exponentially increase in shorter periods, which indicates that droughts occur often in the region with reference to all of the *SPI* and *SPEI* timescales. Conversely, in comparison to Lake Marmara, the *SPI* and *SPEI* drought curves for Salihli station, which is a relatively wetter location, reveal a decreasing risk of drought considering the slopes of the probability curves (Figure 9b,d).

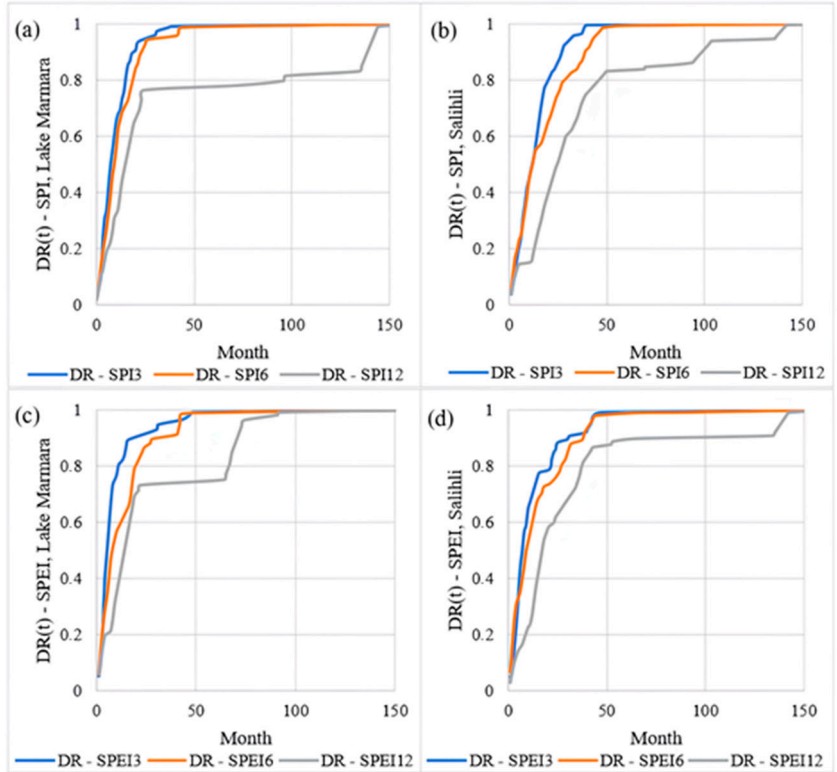

**Figure 9.** Drought risk curves of Lake Marmara (Golmarmara) and Salihli station according to (**a**,**b**) *SPI* and (**c**,**d**) *SPEI*.

### 3.3. Discussion

The presented study adapts the Kaplan–Meier survivability estimator method for the purpose of regional drought risk analysis. There are some other studies on the subject, where the researchers estimate drought risk with other statistical methods, such as Average Recurrence Interval [52], likelihood of impact occurrence [3], or the correlation between drought indices and their impacts [53], examining the time series of a single or several drought index/indices, or as a combination of drought hazard and vulnerability indices [54]. Although most of these studies involve complex calculation processes with rich data requirements, the results represent the related study areas at a coarser resolution [55,56].

However, the presented study suggests a simpler but more efficient approach. First, the proposed methodology enables the easy computation of probabilities, and can be applied to various drought indices. The previous studies examining agricultural drought risk using *SPI* alone also have the potential to underestimate the agricultural drought risk under conditions of global climate change [56–58]. Consequently, the study has potential applicability as a flexible tool that can be used in diverse studies. Second, the drought risk maps generated, which can be produced in high resolution (<30 m) depending on the data availability, are emphasized for their substantial value to stakeholders. Owing to their accessible and interpretable nature, both technical and non-technical stakeholders can comprehend and evaluate the provided maps. This raises their value, as conflicts among stakeholders are often rooted in a lack of comprehension of technical nuances, which may

not be immediately apparent to non-technical counterparts. Furthermore, high-resolution maps can also provide more information on drought risk in the plots of individual farmers.

The utility of drought risk maps extends to decision-makers, aiding in the identification of high-risk areas within the examined region. This information, when demonstrated in terms of spatial and temporal considerations, enables the formulation of targeted preventive measures, including but not limited to local irrigation rotations, adjustments to crop patterns, and investments in irrigation infrastructure. The applicability of such comparative analyses extends not only within the studied region, but also to neighboring areas, reaching from river basins to national and international boundaries.

Moreover, the study underscores the value of drought curves when used in enhancing the understanding of localized drought risk trends. These curves offer a comprehensive perspective compared to drought risk maps, particularly for focal points of significance and hotspots like degraded wetlands.

The specific outcomes of the study are as follows: firstly, we observed an increase in drought risk with the extension of the inquiry period, which is attributed to the inclusion of more months during typically dry periods; secondly, we saw a decrease in drought risk as the timescale of the *SPI* or *SPEI* calculations increased, owing to the cumulative effects that tend to average out extremes; thirdly, we found consistent results in the *SPI* and *SPEI* drought risk maps plotted across most of the study area, with slight variations attributed to the incorporation of potential evapotranspiration (*PET*) by *SPEI*; and finally, we identified high-drought-risk areas through drought curves characterized by steeper initial slopes.

## 4. Summary and Conclusions

The integration of the Kaplan–Meier estimator method into drought risk assessment yields practical and visually informative tools for a diverse array of decision-makers and stakeholders, ranging from individual farmers to larger-scale farming enterprises, and governmental bodies and municipalities.

This tool offers easy probability calculations and can be applied to various drought indices, making it a flexible tool. Importantly, the drought risk maps generated are valuable for stakeholders to use due to their clear and accessible nature, bridging the gap between technical and non-technical understanding. These spatially distributed drought risk maps also aid decision-makers and water-managers in exposing more vulnerable areas of the river basin based on historical evidence, enabling targeted interventions like irrigation management and infrastructure investment. Additionally, the study highlights the value of drought curves in relation to understanding localized trends, providing a deeper insight into specific hotspots like degraded wetlands. The low data requirement of the selected drought indicators, the method's adaptability to other drought indicators, and the easy-to-understand visual representation of the results make this adaptation promising, and extend its applicability beyond the studied region, reaching into neighboring areas and potentially influencing national and international water management strategies.

On the other hand, it must be also underlined that the proposed methodology has its limitations, mostly due to data availability and the accuracy of the fictitious values that are extrapolated between stations. The *SPI/SPEI* timescales utilized (3-, 6-, and 12-month) while assessing the drought risks can be optimized in future studies to focus on a specific aim, supported by the in-situ measurement or records of historical droughts, such as detecting short-term extreme droughts, which may be unobservable due to the accumulation of indicator values when using longer timescales. The same methodology can be applied over longer timescales for detecting longer, more persistent droughts with lower magnitudes. Moreover, the characteristics of the drought curves, such as the slope or angle of the initial section of the curve, etc., can be classified into post-defined sub-categories such as wet and dry states.

**Supplementary Materials:** The following supporting information can be downloaded at: https://www.mdpi.com/article/10.3390/agriculture14030503/s1, Table S1: Data availability of the selected 27 meteorological stations; Tables S2–S217: Kaplan–Meier estimator, drought risk calculations for all 27 meteorological stations.

**Author Contributions:** Conceptualization, C.P.C. and M.C.G.; methodology, C.P.C. and M.C.G.; software, C.P.C. and M.C.G.; validation, C.P.C. and M.C.G.; formal analysis, C.P.C. and M.C.G.; writing—original draft preparation, C.P.C. and M.C.G.; writing—review and editing, C.P.C. and M.C.G.; visualization, C.P.C. and M.C.G.; supervision, C.P.C.; project administration, C.P.C.; funding acquisition, C.P.C. All authors have read and agreed to the published version of the manuscript.

**Funding:** This research was carried out within the MARA-MEDITERRA project, which received funding from the Horizon 2020 European Union Funding for Research & Innovation under the Partnership for Research and Innovation in the Mediterranean Area Programme (PRIMA) Grant Agreement No. 2121 and its sister project UNIMED that has received funding from TUBITAK-MCST Joint R&I Call, which is also supported by PRIMA. The APC was funded by the MARA-MEDITERRA Project (grant No. [2121] [Mara-Mediterra] [Call 2021 Section 1 Water RIA]).

**Institutional Review Board Statement:** Not applicable.

**Data Availability Statement:** The data presented in this study are available on request from the corresponding author.

**Conflicts of Interest:** The authors declare no conflicts of interest.

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
