# Peer review of "Meteorological and Agricultural Drought Risk Assessment via Kaplan–Meier Survivability Estimator"

_agriculture, doi:10.3390/agriculture14030503_

Round 1

Reviewer 1 Report

Comments and Suggestions for Authors

The manuscript introduced a new method called the ‘Kaplan-12 Meier Estimator‘, which is usually used in medicine, to assess agricultural drought. The idea is quite interesting, but there remained some questions before the manuscript can be accepted.

Major:

1. The manuscript was submitted to ‘Agriculture’, also drought is quite important in agriculture, the manuscript should mention more relationship about Drought and Agriculture.

The two index SPI, SPEI was meteorological drought index, what is the shortcomings of these two indexes when assess agriculture drought? What is the advantage of the newly introduced ‘Kaplan-12 Meier Estimator’ when assess agriculture drought?

2. Results and Discussion: Traditionally, the two index SPI and SPEI had its own classification scale as the author stated in Table 1. What is the different when assess drought using the old method and the newly introduced method, and why? What are the implications when using the new method to decision makers? A comparison should be stated.

3. Results and Discussion: sub-title is needed.

Minor:

1. line 12 "Kaplan-Meier Estimator,"‘Kaplan-Meier Estimator‘,

2. line 88 introduce some the agriculture conditions of the GRB

3. line 177 what is the spatial interpolation

Comments on the Quality of English Language

minor editing is needed

Author Response

Dear Reviewer 1, please see the attachment for our point-by-point responses to your valuable comments.

Reviewer 2 Report

Comments and Suggestions for Authors

The article is titled 'Meteorological and Agricultural Drought Risk Assessment via Kaplan-Meier Survivability Estimator'. The study aims to leverage drought risk assessment by treating past droughts as "events." As part of the case study, the authors used the SPI and SPEI indicator methods at different time steps (3, 6 and 12 months) for 27 meteorological stations in the Gediz River basin, located in western Turkey. Based on the SPI and SPEi index, maps were prepared on various time scales.

The article is interesting, but in my opinion it requires slight improvement:

- the adopted methodology requires explanation. While the method of calculating the SPI and SPEI values has been explained, there is no explanation of the definition of risk and no more information about the estimator itself and an indication of its main assumptions. Please define the definition of drought risk. What drought parameters were presented: drought duration, intensity, or other parameters? I miss a short description of these parameters. What method were the maps prepared (interpolation)? I suggest presenting the methodological part in the form of a diagram

- the introductory and discussion parts need to be supplemented with more references. Emphasize the importance of the methods used, their advantages and disadvantages

Technical notes:

- weather stations marked with triangles (light green) may be confused with the map background, I suggest changing the color

​

Author Response

Dear Reviewer 2, please see the attachment for our point-by-point responses to your valuable comments.

Reviewer 3 Report

Comments and Suggestions for Authors

The study aims to apply the Kaplan-Meier Survivability Estimator to assess meteorological and agricultural drought risk via the Standardized Precipitation Index (SPI)  and the Standardized Precipitation Evapotranspiration Index (SPEI), respectively, in Gediz River Basin, in Turkey.

This is an interesting study. However, several issues should be resolved (see the comments in the attached manuscript) such as the data availability and the definition of the reference period used, better description of the procedure of estimating the Kaplan-Meier Survivability Estimator and its results, and focus on the comparison of the two drought indices. 

Comments on the Quality of English Language

Moderate editing of English language required

Author Response

Dear Reviewer 3, please see the attachment for our point-by-point responses to your valuable comments.

Round 2

Reviewer 1 Report

Comments and Suggestions for Authors

The questions were well answered and the manuscript was much improved.

Comments on the Quality of English Language

none

Reviewer 3 Report

Comments and Suggestions for Authors

The revised manuscript is improved as a result of the adequate author responses to the issues and concerns raised. Therefore, I recommend its publication

Comments on the Quality of English Language

Minor editing of English language required